# Is an Ambulatory Biofeedback Device More Effective than Instructing Partial Weight-Bearing Using a Bathroom Scale? Results of a Randomized Controlled Trial with Healthy Subjects

**DOI:** 10.3390/s24196443

**Published:** 2024-10-05

**Authors:** Tobias Peter Merkle, Nina Hofmann, Christian Knop, Tomas Da Silva

**Affiliations:** Department of Trauma Surgery and Orthopaedics, Klinikum Stuttgart-Katharinenhospital, Kriegsbergstraße 60, 70174 Stuttgart, Germany; nina.hofmann00@outlook.com (N.H.); c.knop@klinikum-stuttgart.de (C.K.); t.dasilva@klinikum-stuttgart.de (T.D.S.)

**Keywords:** evidence-based medicine, biofeedback, orthosis, partial weight-bearing, pressure sensor, rehabilitation

## Abstract

So far, there have been no high-quality studies examining the efficacy of outpatient biofeedback devices in cases of prescribed partial weight-bearing, such as after surgery on the lower limbs. This study aimed to assess whether a biofeedback device is more effective than using a personal scale. Two groups of healthy individuals wearing an insole orthosis were trained to achieve partial loading in a three-point gait within a target zone of 15–30 kg during overground walking and going up and down stairs. The treatment group (20 women and 22 men) received continuous biofeedback, while the control group (26 women and 16 men) received no information. Findings were compared in a randomized controlled trial. Compliance with partial loading without biofeedback was poor; on level ground and stairs, only one in two steps fell within the target area, and overloading occurred on at least one in three steps. The treatment group reduced the percentage of steps taken in the overload zone to ≤8.4% (*p* < 0.001 across all three courses) and achieved more than two-thirds of their steps within the target zone (*p* < 0.001 on level ground, *p* = 0.008 upstairs, and *p* = 0.028 downstairs). In contrast, the control group did not demonstrate any significant differences in the target zone (*p* = 0.571 on level ground, *p* = 0.332 upstairs, and *p* = 0.392 downstairs). In terms of maintaining partial load, outpatient biofeedback systems outperform bathroom scales.

## 1. Introduction

Currently, partial weight-bearing of 20 kg and immobilization are the most commonly applied postoperative rehabilitation principles in the early stages following ankle joint surgery [1]. This focus on partial weight-bearing primarily aims to prevent complications related to wound healing and mitigate the risk of secondary loss of reduction [2,3]. However, this strategy also carries potential risks, including osteopenia, muscle atrophy, joint stiffness, and deep venous thrombosis, making some degree of weight-bearing necessary [3,4,5,6]. In light of the potential risks associated with postoperative rehabilitation protocols, the necessity for more precise methods for assessing gait parameters and monitoring progress in rehabilitation becomes particularly evident.

An option for assessing gait parameters and tracking disease progression in rehabilitation settings is the use of three-dimensional motion analysis, which significantly contributes to improved outcomes [7,8]. High-precision force plates and motion-capture cameras are utilized to measure ground reaction forces. However, such gait assessments are limited to a select few centers due to their high costs, space requirements, and staffing needs [9]. Furthermore, force plates are limited to recognizing a single step and may not accurately reflect real-life outpatient circumstances, as laboratory conditions can influence test subjects’ behavior [9]. Additionally, monitoring patients’ weight-bearing compliance in a home and active functional environment presents significant challenges, as assessing adherence to the prescribed rehabilitation protocol can be difficult [10]. Finally, monitoring patients’ weight-bearing compliance in a home and active functional environment presents a significant challenge, as it is difficult to assess their adherence to the prescribed rehabilitation protocol [11,12]. In recent years, there has been an increase in the development and research of mobile health technologies outside traditional motion-capture laboratories. As a direct result of technological advancements, ambulatory insole measurement devices are now commercially available, enabling the analysis of objective weight forces. These devices are portable, less expensive, and user-friendly, allowing for easy quantification of ground reaction forces in everyday environments such as clinics and homes. Since clinicians are generally more interested in vertical component reaction forces, these devices typically focus solely on that aspect. Consequently, the number of studies focusing on these ambulatory insole devices has increased [11,12,13,14,15,16,17,18,19]. Research has also addressed subject compliance. Studies indicate that training and adherence to specific partial weight-bearing using a discontinuous measuring method, such as a bathroom scale, often result in poor compliance with a limited weight-bearing regime. Typically, patients receive instructions from a physiotherapist on how to perform partial weight-bearing on crutches using a bathroom scale before being discharged from the hospital after surgery. However, patients often struggle to adhere to these guidelines due to a lack of continuous feedback after stepping off the scale [15,20,21,22]. In our preliminary studies, we demonstrated that patients with an ankle fracture deviated significantly from their clinical recommendations. Only one-third of the steps fell within the designated target zone, while every fifth step exceeded the recommended weight-bearing, and every second step fell below it. Another important finding regarding compliance was that a quarter of the patients did not bear any weight on their legs before discharge. This is where ambulatory insole devices prove beneficial, as they provide patients with real-time feedback at each step, ensuring adherence to the prescribed partial weight-bearing. In our study, we significantly increased the number of steps within the target zone and reduced the number of steps in the overload zone to just 10% [23].

To date, no high-quality studies have been published regarding the effectiveness of ambulatory biofeedback devices. A notable lack of randomized controlled studies exists to demonstrate the significant superiority of ambulatory insole devices over the intermittent measurement method provided by a bathroom scale. The current level of evidence is low, primarily due to a limited number of cases, the lack of randomization, and a heterogeneous sample of participants. The aim of this study was to determine whether an ambulatory biofeedback insole device is more effective than using a bathroom scale. We hypothesized that implementing a biofeedback system would improve weight-bearing compliance compared to a standard bathroom scale. This study was grounded in recommendations from our professional association’s evidence-based medicine group, raising the level of evidence to Ib according to the Oxford Centre for Evidence-Based Medicine (OCEBM) Levels of Evidence [24,25].

## 2. Materials and Methods

### 2.1. Statistical Analysis

Prior to this study, a power analysis was conducted by the Biometrics Department of the University of Tuebingen to determine the necessary sample size. Targeting a power of 80% and a significance level of *p* < 0.05, the required number of participants for each subgroup was estimated at 42, resulting in a total of 84 healthy participants. The normality of the data distribution was assessed using the Shapiro–Wilk test, which confirmed that the data were not normally distributed. Due to this violation of normality, we applied the Wilcoxon Signed-Rank Test as a non-parametric alternative to the *t*-test, allowing us to compare the effects of biofeedback within both the treatment and control groups. Data are reported as means and standard deviations. The Mann–Whitney U test was employed for comparisons between the groups.

### 2.2. Inclusion and Exclusion Criteria

Strict selection criteria were employed to ensure a homogeneous sample population. Inclusion criteria required participants to have sufficient upper-body coordination and strength to perform partial weight-bearing with crutches, with a minimum age of 18 and a maximum age of 60 years. Participants with a history of joint or bone injuries were excluded, as were those unable to complete the exercise sequence for various reasons. Factors contributing to exclusion included a lack of upper- and lower-body coordination and strength, as well as medical disorders and comorbidities such as heart failure, poor pulmonary function, or neurological symptoms. A history of cognitive issues was also a disqualifying factor.

Additional exclusion criteria included any type of lower foot malformations, acute or chronic injuries or diseases, gait disorders due to muscle weakness, shoe sizes outside the measurement range (EU size 36–46), and individuals who were either underweight (BMI < 18.5 kg/m²) or overweight (BMI > 30.0 kg/m²).

### 2.3. Subject Characteristics

The sample examined consisted of 84 healthy individuals (ASA I; 46 women and 38 men) with an average age of 28.7 years (range: 18 to 56 years). There were no dropouts. The American Society of Anesthesiologists (ASA) classifies physical status, with Class I denoting normal health, indicating that individuals are healthy, non-smokers, and have no or minimal alcohol use [26]. The average body mass index (BMI) of participants was 23.5 kg/m² (range: 19.7 to 29.4 kg/m²), and the median shoe size was 41 (range: 37 to 46 EU). After randomization, the control group included 20 women and 22 men with an average age of 30.4 years (range: 19 to 56 years) and an average BMI of 23.0 kg/m² (range: 19.9 to 29.4 kg/m²). In this group, the left leg was offloaded 22 times (14 times for females and 8 times for males), while the right leg was offloaded 20 times (12 times for females and 8 times for males). The treatment group consisted of 26 women and 16 men with an average age of 27 years (range: 18 to 56 years) and an average BMI of 24.0 kg/m² (range: 19.7 to 29.4 kg/m²). In the treatment group, the left leg was offloaded 21 times (9 females and 12 males), with the right leg also offloaded 21 times (11 females and 10 males).

### 2.4. Measuring System

For this study, a commercially available system (SP Air Smart Walker, Sporlastic, Nuertingen, Germany) was selected. This lower leg foot orthosis was customized to each subject’s foot size and was used to detect continuous real-time weight-bearing status. The in-orthosis body weight measuring system consists of two main components: a flexible, ergonomically shaped, single-use force-measuring sole (Sens2Go, Golex AG, Basel, Switzerland), which is inserted into the orthosis, and a reusable miniature control unit (Sens2Go, Golex AG, Basel, Switzerland).

When weight is loaded on the lower limb during gait, pressure increases on the four pressure sensors integrated into the insole. The sensereader, attached to the orthosis, connects to the insole and functions as a data logger, assigning a maximum weight load to each step. It can be removed for charging via USB and includes a Bluetooth transmitter and receiver. Once set up, it can operate independently of a mobile phone. A Bluetooth connection through an app allows for the downloading of raw data. In addition to training applications, the device can be utilized for research purposes, generating cumulative evaluations that assess prior exposure levels without collecting personal data. An audiovisual signal can be activated or deactivated via the software; when turned off, the device functions solely as a measuring instrument. When activated, it provides live feedback through an audiovisual signal generated by the sensereader once the participant achieves the set value of 20 kg while walking (Figure 1). The manufacturer specifies that the variability in measurement findings of the force-measuring sole is ≤1%. The sampling frequency of the system is 50 Hertz. The device is a CE-marked medical device and can be rented for a low three-digit fee for a duration of six weeks. The system was provided pre-calibrated by the company.

### 2.5. Experimental Protocol

The study design is presented in Figure 2. The examination for each individual was conducted in a single visit. All participants were trained to perform a three-point gait using forearm crutches and instructed to engage in partial weight-bearing with 20 kg, targeting a weight range between 15 kg and 30 kg, as measured by a bathroom scale. The leg designated for partial loading was determined in advance through randomization.

Participants were then asked to walk a set distance of 50 m on level ground and to climb and descend two levels (22 stairs) at their preferred speed immediately following this procedure. In the first cycle, baseline recordings were conducted without biofeedback prior to group formation. Following this, randomization was implemented to assign participants to two groups: 42 individuals received audiovisual biofeedback (treatment group), while an equal number received no biofeedback (control group).

The second cycle was conducted in the same manner as the first. After a brief break to prevent physical fatigue, participants completed the course again. There was no additional practice with weight-bearing on the bathroom scale, and the same leg was unloaded during both runs. To avoid any post-response learning, participants were not informed of their results after completing their runs.

The peak force [N] for each individual step was recorded. The total maximum loads were categorized into three load ranges (Figure 3). The range below 15 kg was classified as underloaded, the range above 30 kg was classified as overloaded, and the range between 15 and 30 kg was defined as the target zone.

## 3. Results

Table 1, Table 2 and Table 3 display the three zones—underload, target zone, and overload—in each row. The columns represent the activities: level walking, stair ascent, and stair descent. The tables indicate the percentage of steps that fall into each respective zone.

Table 1 presents the load results of the first cycle of all 84 test participants after learning partial weight-bearing on a scale.

According to the data, approximately half of the stair steps were taken within the target zone (51.6% while ascending and 52.4% while descending), with nearly a third being overloaded (36.0% and 32.0%, respectively). On level ground, 45.1% of the steps were in the overload zone, while only 42.5% were in the target zone. No significant differences were observed between sides or genders in any zone during the initial cycle (each *p* > 0.05).

Table 2 presents the results for the control group during the second cycle.

The findings indicated that there was a shift in load towards the overload zone, which became significant (*p* = 0.045) even during stair descent.

Table 3 presents the results for the treatment group during the second cycle.

The data revealed that the use of biofeedback resulted in a transfer of loads from the overload zone to both the target zone and the underload zone. It was possible to achieve a reduction in the total number of steps carrying excessive weight (>30 kg) to 8.4% or less.

In the control group during the second cycle, women had more steps in the target zone than men (*p* = 0.038). Otherwise, no significant differences were observed between genders within the control and treatment groups across any zone (each *p* > 0.05). No differences were noted between offloading the right and left legs in any other zone for both groups (*p* > 0.05).

## 4. Discussion

The findings of this study lead to two important conclusions. First, the use of a bathroom scale is inadequate for individuals to achieve the prescribed partial weight-bearing. Second, the use of an outpatient biofeedback device results in a significant improvement in achieving a specified partial load.

Previous studies have also observed improvements in partial weight-bearing performance through feedback compared to conventional teaching methods used in clinical practice, such as verbal instructions or bathroom scales. Hershko et al. compared a control group (without biofeedback) to an intervention group (with biofeedback) and found that the intervention group benefited significantly from the biofeedback (*p* = 0.011), while the control group showed no improvement [28]. Similarly, in the study by Hurkmans et al., the difference between the biofeedback group and the non-biofeedback group was also significant (*p* = 0.035) [29]. However, despite these promising findings, the overall quality of evidence in other studies remains low due to methodological limitations such as small sample sizes, heterogeneous participant groups, and the lack of a randomized controlled design. For instance, in the study by Hershko et al., the evaluation involved 36 patients with a wide variety of conditions, such as total hip arthroplasty, hip intramedullary nailing, tibial plateau surgery, hip hemiarthroplasty, hip nailing, and acetabular surgery. Furthermore, these patients were subject to varying postoperative rehabilitation protocols, such as toe touch versus partial weight-bearing [28]. Similarly, Hurkman initiated his study with 66 patients but encountered a significant dropout rate of 42.4%, which further complicates the interpretation of the results [30].

In light of these findings, our study built upon these insights to specifically address the load distribution during different activities. This was the first study to demonstrate these results using a prospective randomized controlled study design. Only a maximum of every second step could be placed in the target zone in the first cycle, while every third step was overloaded. During the second cycle, there was even a further deterioration in the load in the overload zone. With biofeedback activated, we achieved a significant reduction in the number of steps in the overload zone on level ground (*p* < 0.001), decreasing from 45.1% to 6.6%. On the stairs, we achieved a reduction to less than one out of every ten steps (*p* < 0.001).

To comprehensively evaluate load management, it became essential to include an underload zone along with the target and overload zones. This study is based on a subjective classification of a three-zone separation—underload, target, and overload zones—for which we established specific definitions. These zones have been adapted to align with the medical processes in our department. The target zone principle is already common in clinical practice. Rather than solely focusing on loads above or below 20 kg [17,18], several authors have addressed the concept of a target load, emphasizing the need to avoid both overexposure and underexposure [18,22,23,28,29]. For this purpose, we conducted training exercises with the subjects aimed at a target load of 20 kg. This concept is based on Warren’s description of the overshoot phenomenon (Figure 3), which resulted in a one-third increase in the size of the target zone above the biofeedback signal in this study and has been previously utilized [27,28,29]. However, in clinical practice, there has never been a conclusive answer to the question of what the ideal therapeutic range should be, and a target zone has not yet been defined for any entity. Future researchers may adapt the variables used in this study to their specific situations. The zones were used solely as a baseline for comparing the groups in this study.

Despite these insights into load management, there remains a notable gap in the adoption of mobile biofeedback devices in clinical settings. This is partly due to the high manufacturing and maintenance costs associated with these devices. Additionally, cost-effectiveness analyses of the devices are lacking, as noted in Jagtenberg et al.’s summary [3]. Although the systems used in this study were less expensive than those from other manufacturers, health insurance providers refused to cover the costs of the devices due to the absence of a prospective randomized study prior to approving any treatment for patients. This requirement is reasonable, as previous studies have not exceeded Level IV evidence, prompting the investigation in this study.

Moreover, an ongoing question among researchers pertains to the long-term effectiveness of unloading with the biofeedback system. They wonder whether the therapeutic benefits are maintained even if the system is not used continuously throughout the entire treatment period or whether consistent application over the typical six weeks is necessary to achieve the desired unloading. Vasarheli et al. trained both subjects and patients using a weighing machine and 200 N partial weight-bearing while walking in a three-point gait. Measurements were taken over three consecutive days, and it was found that neither healthy volunteers nor patients were able to perform the prescribed partial weight-bearing of 200 N during the dynamic measurement on all three test days; all participants loaded too much [2]. Hershko et al. compared a control group without biofeedback to an intervention group with biofeedback over ten days. While the control group showed no improvement, the intervention group benefited significantly from daily use of their insole device during the first five days and demonstrated a positive effect even after an additional five days following the removal of the insole device. The effect remained significant during this period despite a successive increase in loads [28]. Hurkmans et al. observed a similar effect in their study. In their follow-up after seven days, the number of loads in the intervention group increased, and despite prior daily biofeedback training, there was no longer a significant difference compared to the control group after just one day. After an additional three weeks without further training, the number of loads continued to increase in both groups, with no detectable difference [29]. Short-term retention effects do exist; however, they are only observable for a very brief period, typically lasting just a few days. No studies to date have demonstrated longer-lasting retention. The removal of feedback resembles a discontinuous method, and we believe that the full benefits of biofeedback can only be realized through continuous application.

### 4.1. Limitations

In this study, the effectiveness of the device was evaluated using a group of healthy participants rather than patients. Additionally, the criteria for inclusion and exclusion were carefully selected to create a homogeneous research group, as patients exhibit varying degrees of performance [23,31]. It can be assumed that patients in pain or with significant swelling may not load at all or may do so infrequently. Factors such as lower limb soreness, edema, and decreased joint mobility negatively impact the ability to walk freely and without restrictions [23,31,32].

While focusing on healthy participants facilitated a controlled evaluation of the device’s effectiveness, it is important to note that we employed a combined audiovisual biofeedback approach without incorporating haptic feedback. Consequently, we were unable to determine which type of feedback provided the most benefit. Generally, young healthy subjects handle the audiovisual biofeedback well; however, we lack information on how individuals with visual or auditory impairments might respond. Furthermore, both forms of feedback depend on environmental conditions. Since the study was conducted indoors, the visibility and clarity of the information might be compromised in bright or distracting environments. Similarly, noisy conditions can hinder the clarity of acoustic feedback, potentially diminishing its effectiveness.

To advance the discussion regarding the device’s effectiveness, it is crucial to address the limitations of the studied population. Testing was limited to a homogeneous group of healthy individuals, raising questions about the reactions of different patient groups, particularly those with sensory impairments. Jagtenberg et al. suggested that understanding the practicality and therapeutic validity of existing biofeedback training devices is essential for optimizing exercise instructions for patients [3]. We concur with this finding. Gaining insights into the feasibility and clinical validity of study protocols is critical for optimizing weight-bearing instructions. Therefore, we believe that evaluating these devices on healthy participants is essential for understanding physiological three-point gait and the deviations that lead to pathological gait patterns. The insights gained here can serve as a foundation for future research involving patients.

### 4.2. Future Work

With the biofeedback insole device used in this study, it is possible to obtain objective measurements over extended periods. This capability not only allows for assessing patient compliance in the home environment but also enables effective load management. Such data can facilitate the development of patient-specific rehabilitation measures tailored to factors such as age, gender, and the nature of an injury.

## 5. Conclusions

When using an audiovisual biofeedback device, study participants were able to more accurately reach the target loading zone compared to when they did not use the device. Therefore, utilizing a biofeedback insole device could be a promising approach to help ensure that individuals wearing a walking brace do not overload or underload their feet. Patient-specific rehabilitation may benefit from its broad application in the home setting.

## Figures and Tables

**Figure 1 sensors-24-06443-f001:**
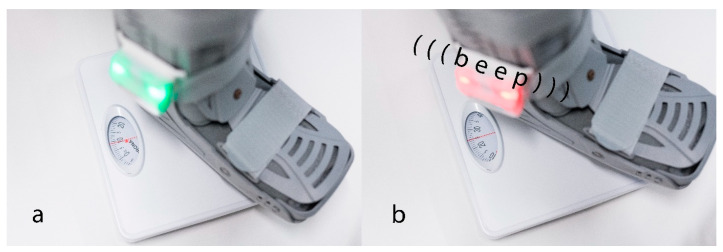
The device outputs loads under 20 kg with a green light (**a**) and loads over 20 kg with a red light (**b**), accompanied by an audible alarm sound. The removable sensereader is located on the outside of the orthosis. This makes it easy to see and ensures that it does not get in the way while walking.

**Figure 2 sensors-24-06443-f002:**
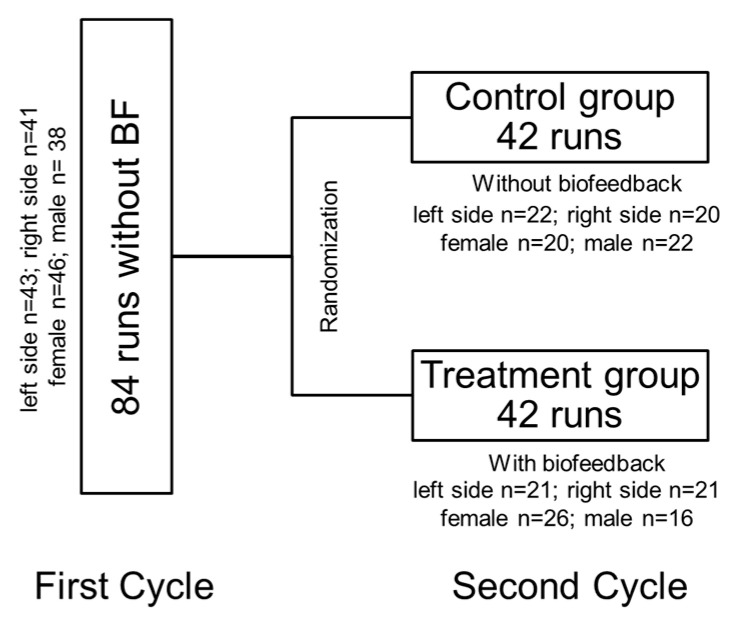
Diagram of the study’s design (BF = biofeedback).

**Figure 3 sensors-24-06443-f003:**
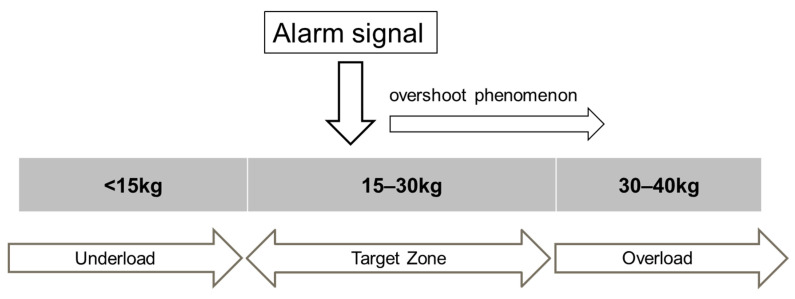
This classification was determined by the authors. It is based on Warren’s preliminary research where patients continue to exert a load after an alarm is triggered due to a physiological response time of 150 to 250 milliseconds [27,28]. This response leads to excess weight on the injured limb. To address this, it is recommended to set the alarm slightly below the maximum weight-bearing limit to account for this delay and additional loading.

**Table 1 sensors-24-06443-t001:** The first cycle included all participants without biofeedback. All loads were divided into three categories: underload (<15 kg), target zone (15–30 kg), and overload (>30 kg). The sum of the steps per load zone is given in %. The mean step count and its standard deviation are presented in parentheses.

First Cycle n = 84	Underload <15 kg	Target Zone 15–30 kg	Overload >30 kg
Level ground	12.4% (5.7 SD 8.6)	42.5% (20.1 SD 12.8)	45.1% (22.6 SD 18.3)
Upstairs	12.4% (2.8 SD 3.9)	51.6% (12.1 SD 7.2)	36.0% (8.4 SD 7.6)
Downstairs	15.6% (3.5 SD 3.8)	52.4% (12.1 SD 6.2)	32.0% (7.4 SD 6.9)

**Table 2 sensors-24-06443-t002:** Analysis of the control group (no biofeedback). The sum of the steps per load zone is given in %. The mean step count and its standard deviation are presented in parentheses.

Second Cycle Control Group (n = 42)	Underload <15 kg	Target Zone 15–30 kg	Overload >30 kg
Level Ground	10.7% (4.8 SD 6.0) ^a^	39.9% (19.6 SD 13.6) ^b^	49.4% (25.3 SD 19.1) ^c^
(a) *p* = 0.571; (b) *p* = 0.582; (c) *p* = 0.204			
Upstairs	9.1% (2.1 SD 2.9) ^a^	48.0% (11.3 SD 7.7) ^b^	42.9% (10.0 SD 7.6) ^c^
(a) *p* = 0.332; (b) *p* = 0.408; (c) *p* = 0.334			
Downstairs	14.9% (3.4 SD 4.3) ^a^	47.2% (11.0 SD 6.5) ^b^	37.9% (8.8 SD 6.7) ^c^
(a) *p* = 0.392; (b) *p* = 0.139; (c) *p* = 0.045			

**Table 3 sensors-24-06443-t003:** Analysis of the treatment group (with biofeedback). The sum of the steps per load zone is given in %. The mean step count and its standard deviation are presented in parentheses.

Second Cycle Treatment Group (n = 42)	Underload <15 kg	Target Zone 15–30 kg	Overload >30 kg
Level Ground	22.0% (11.0 SD 8.9) ^a^	71.4% (39.8 SD 18.8) ^b^	6.6% (3.3 SD 4.3) ^c^
(a) *p* = 0.017; (b) *p* < 0.001; (c) *p* < 0.001			
Upstairs	19.4% (4.4 SD 4.1) ^a^	72.2% (16.7 SD 5.7) ^b^	8.4% (1.9 SD 2.4) ^c^
(a) *p* = 0.091; (b) *p* = 0.008; (c) *p* < 0.001			
Downstairs	26.6% (6.0 SD 4.2) ^a^	69.2% (15.6 SD 4.5) ^b^	4.2% (1.0 SD 1.3) ^c^
(a) *p* = 0.022; (b) *p* = 0.028; (b) *p* < 0.001			

## Data Availability

We are willing to make the raw research data available for further exchange. Requests can be made directly to the corresponding author.

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
