# Peer review of "Is an Ambulatory Biofeedback Device More Effective than Instructing Partial Weight-Bearing Using a Bathroom Scale? Results of a Randomized Controlled Trial with Healthy Subjects"

_sensors, 2024, doi:10.3390/s24196443_

Round 1
Reviewer 1 Report
Comments and Suggestions for Authors
Although the paper presents interesting content, the introduction lacks sufficient background information, making it challenging to support the main contribution effectively. Furthermore, Section 3 and 4 require a more in-depth academic analysis to strengthen the overall argument.
Reviewer 2 Report
Comments and Suggestions for Authors
Manuscript Number: sensors-3175631
Title: Are ambulant biofeedback devices more effective than the traditional method of teaching partial weight bearing using a bathroom scale? Results of a randomized controlled gait analysis
General Comments
The aim of this study was to determe whether a biofeedback device is more effective than traditional intermittent training therapy, even when considering a randomized controlled study design. The authors hypothesized that dynamic biofeedback training would improve the immediate weight bearing retention compared to conventional scale training. The authors reported that the greater relative number of steps is within a desired loading window after instruction with compared to without biofeedback. This is a well written manuscript. However, there are several fundamental aspects of the methodology and interpretation of the study that require clarification and additional information.
1. Title: ‘…ambulant biofeedback devices…” this phrasing implies that several devices were tested. Please us a more concise title; Teaching -> instructing; the gait analysis is not randomized. This is a randomized trial. Please also add that this study involved healthy subjects.
2. Line 34: that deal with – this is lay language. The manuscript should be copy-edited by a scientific writer.
3. Line 38: a specified partial… The noun is missing here.
4. Line 44-48: Please add that this was done in healthy subjects.
5. Line 50-54: The authors describe their sample size estimation. The rationale for the assumptions underlying these calculations should be provided. Please provide citations to relevant literature.
6. Line 61: such as heart failure. The “as” is missing.
7. Line 63: The authors excluded subjects with any time of lower limb malformations. Does this include conditions such as hallux valgus? How were malformations defined and how was this criterion checked? Radiographically? Visually? Self-reported?
8. Line 65: BMI – missing unit.
9. Line 69: Shoe size would be better reported as median and range.
10. Line 67-73: Were the subjects recruited in a block randomization manner to ensure equal number of men and women in both groups?
Please provide a study flow diagram. How many subjects were screened and excluded for what reasons?
Were there any drop outs?
11. Please provide additional data on the system used. What is the accuracy and reliability of the load parameters? What were the technical specifications (i.e., sensor location, sampling rate, type of data exported)? Calibration procedure? What data was exported? What post processing was conducted? Please provide time series of data – if not in the manuscript then at least in the supplementary material.
12. Line 99-110: Please specify that each subject was assessed in a single visit.
It is unclear why the subjects were instructed on weight bearing of 20kg when the desired loading zone was 15-30 kg.
Please also define better what is later used as first and second cycle / run. Please be consistent.
Was leg dominance assessed? Unloading the dominant or the nondominant leg might result in different results.
13. Figure 2: Define BF in the figure caption.
The use of runs is misleading. 84 subjects completed the first cycle without BF, 42 subjects completed the second cycle for each with or without BF.
14. Line 113: Peak force would be measured in N and not in kg. Please use the correct terminology.
15. Methods: The authors must define the outcome parameter and how this was obtained. This is completely unclear in the current version of the manuscript.
16. Line 120-123: Besides not reporting the target outcome, the authors use t-tests to test their hypothesis. It seems that the outcome is the relative number of steps in one of the three loading categories. For each subject, activity and cycle, the relative number of steps in all three categories ads up to 100%, hence the outcomes are not independent. Moreover, as data are compared between the first and second cycle, the factor cycle is a within subject factor, while the factor group (with or without BF) is a between subject factor. Hence, mixed effects models with the within subject cycle and the between subject factor BF should be used to test if there is a significant interaction between cycle and group. Accordingly, the sample size estimation should have been made for such a model. Moreover, the data must be checked for normality. Especially the data for underload appears to be not normally distributed as the mean minus the standard deviation is negative which is not possible as the outcome can only take on values between 0 and 100. The use of nonparametric tests would be warranted in such cases. The authors must consult a biostatistician to revise their sample size estimation and statistical analysis.
17. How long were subjects trained to walk with crutches? 50 m seems to be a quite long distance. Was this performed on a 50-m walking track or did the measurements include periods of standing or turning? How many steps did the subjects take to cover this distance?
18. Figure 3: Why were the unloading range further categorized into <15, 15-20, 20-25, 25-30, 30-40, and >40 kg in the illustration? This does not correspond to the results section.
19. Tables: There is a huge variability in the data. What are potential causes? Did any of the subjects have prior experience with walking on crutches? Was this related to body weight? It is important to consider that 20kg loading corresponds to 40% body mass in light persons and to 25% or less in heavier persons.
20. Lines 162-190: Several statements are lacking references. Please provide appropriate citations for the statements made.
21. Discussion: A major and highly clinically relevant question is whether the unloading with BF is maintained over a longer period of time or if the BF must always be on during the entire unloading period (usually 6 weeks). This aspect should be discussed in the context of the literature in the discussion section.
22. Line 192-201: This section does not describe a limitation and should hence be moved up to the discussion section.
23. Line 207: “They” – what does this refer to?
24. Line 212: What are cross-time comparisons? Do you mean within-subject comparisons or comparisons over time? Pre- to post-intervention comparison would be more appropriate.
25. Conclusions: The conclusions are too general and must be supported by the data and results presented in the manuscript.
Comments on the Quality of English LanguageThere are some grammatical errors and choices of wording that should be revised.
Reviewer 3 Report
Comments and Suggestions for Authors
General
Thank you for the opportunity to read and review this work. After reading this manuscript, I understood the paper to be about how a visual-auditory biofeedback device, linked to a pressure sensing insole, could be used within a walking brace to ensure individuals are better able to stay within a weight bearing target range compared to without.
While the study was interesting to read about and the results support the use of this device, the manuscript needs a lot of work in the writing in order to make it ready for publication. Not only does the structure of the paper need to be re-worked but I think the authors underestimated the influence of proper terminology and vocabulary on the readability of the paper.
Major Comments:
1. While the discussion later touches on this (and I suggest moving it to the introduction), the abstract and introduction are lacking justification as to why this study needs to be done (the gap in the literature that it is addressing is weak). The authors should make clearer the novelty of this work, the target population, and what the implications are. For example, it is not clear why overloading and underloading a walking brace is problematic.
2. The methods were not written in a way that the reader could easily or clearly understand what the protocol was. This needs to be improved so that the results can be understood and properly interpreted.
3. The authors often mentioned and referred to “Conventional teaching”, however, from my understanding this just referred to individuals standing on a standard bathroom scale to understand what 15kg to 30kg loading feels like. In my opinion, this is not teaching, but rather a familiarization session. Teaching to me would imply that a professional or a therapist is showing them how to properly load the leg, and involve multiple session to “learn” how to do it properly. Therefore, this study to me has nothing to do with teaching but rather it is about being able to judge how much load is being applied.
4. The conclusion of this manuscript doesn’t match the results presented. The authors speak about multiple devices compared to conventional teaching (see previous point), economics, comparisons between participants, and the feasibility of different devices. These are not points which the results directly speak to and the authors should be sure to focus on what the actual outcome of their work is. From my understanding the results show that when using a visual-auditory biofeedback device, linked to a pressurized insole mounted in a walking brace, participants could more accurately hit the target loading zone, compared to when they didn’t use the device. Therefore, using a visual-auditory device with a pressurized insole could be promising to help ensure that individuals wearing a walking brace don’t overload or underload their foot.
Minor Comments:
Abstract
Line 10: Sentence is strangely written. I would suggest breaking it into two sentences and to make it clearer what effect you want to evaluate and why.
Line 10: What is the biofeedback for? This could be referring to many things and more precise terminology should be used.
Line 12: What type of people are you referring to? Healthy? What are the two groups of people? You could also include the sample size and sex distribution here.
Line 13: A bit more background as to what “3-point gait” refers to would be beneficial. As the target population also hasn’t been explained it is not clear what type of individuals these (is it a health individual on crutches, is it an individual with an amputation on crutches, etc).
Line 13: The authors write that participants were “taught” but how was the teaching done? What does this refer to? What were participants taught to do (this only makes sense if the reader knows what the goal of the study is).
Line 14: The “T” in treatment should not be capitalized. Please be careful and proofread.
Line 14: Please describe and define what is meant by “continuous feedback”. For example, how long is the feedback for? What modality was used (auditory, visual, haptic, multimodal)? What interface was used?
Line 16: How is “poor” defined? And how was compliance measured?
Line 16: It is hard to understand what is meant by “On ground and on stairs….” Without knowing what the participants had to do?
Line 18: If could be nice if you could report the exact p-value. Typically, the exact value is reported (P = .04) rather than expressing a statement of inequality (P < .05), unless P <.001.
Line 19: Hard to understand and interpret the results without knowing what the participants did.
Line 20: What is a learning device? Is this a specific type of rehabilitation device that is standard in the field? If not, please explain what a “learning device” is.
Line 12: It seems that here you are saying that bathroom scales are often used to achieve a specific goal (avoid overloading) - why not use this earlier to setup the problem statement or explain in the protocol what their role is in this study.
Line 13: Who are the “users” being referred to and and what do they want to compare and why? Intra or inter user comparisons? What does this have to do with a gait lab?
Line 22: “Developing concepts” This is very vague, what concepts are you referring to? This sentence is not very clear.
Introduction
Line 28: I feel like there are two ideas in this sentence - one is about tracking weight bearing, and the other about measuring weight bearing. Please rewrite more clearly, perhaps revisiting the story you are trying to tell in this manuscript.
Line 34: “…studies that deal with various types of devices.” This is very vague, please be more specific. What type of studies? What is their general topic or what do they address? And what are "various types of devices”?
Line 36: I suppose that with "outpatient biofeedback devices" you are referring to the ambulant insole devices? If yes, please make this clearer that this is what it refers to, to improve readability.
Line 36: what is meant by a "teaching method". What are you teaching? To whom? This idea needs to be properly introduced or rephrased as explained in the major comments.
Line 36: This idea of “…already concerned with the compliance of subjects and patients.” needs to be better integrated otherwise it feels very disjointed.
Line 38: What does “…with a specified partial…” mean? Please clarify.
Line 38: “…discontinuous measuring method like a bathroom scale…” instead of just referring to the literature for this, it could help the reader if you explain in lay terms what the current standard is, what the goal is and in what situations it is used. Then the reader can better understand how limitations of discontinuous measuring is and how a bathroom scale fits into all of this
Line 39: “…limited weight bearing regime” you need to consider that your audience may not know what this is, perhaps providing more details or being more explicit could be useful.
Line 23: “controlled randomization” and a “randomized control study” are not the same thing. Please be more intentional with your terminology.
Line 44: How do you define "an appropriate number"? And why homogeneous? I don't feel you have adequately explained why previous studies with heterogeneous samples are undesirable. What does heterogeneous comprise? The reader doesn't even know your target population.
Line 45: “…more effective than traditional intermittent training therapy…” this idea has been insufficiently introduced for the reader to understand what the problem with it is or how it is typically conducted. Therefore, the reader cannot judge the efficacy of a biofeedback device.
Line 45: The use of the term biofeedback device is very vague and could be a number of things.
Line 45: “…even when considering a randomized controlled study design.” This doesn’t mean anything in this context.
Line 48: “Conventional scale training” please be more consistent in your terminology. It feels like the same terminology hasn’t been used twice so it is extremely confusing what you are referring to at each time and if it is the same thing or something different
Line 48: Why is retention important?
Methods
Line 52: Could you explain a bit on what this high SD estimation came from? What is it based on? Can you provide some references?
Line 58: “experienced joint of bone injuries” does this apply to the whole body? Could you please specify this? And perhaps provide an example of what this means (e.g., does having broken your toe at 5 years old exclude you from this study?).
Line 58: you write “were excluded” twice in the same sentence so it sounds repetitive. Please remove one instance of this.
Line 59:”…complete exercise sequence” how was this assessed
Line 62: cognitive such as dementia, or all mental health/psychiatric disorders?
Line 67: “ASA I” abbreviation (and in this case the classification) must be defined
Line 67-73: This section is more results then methods. You could also consider representing this information in a table rather than text to make it more readable
You might also indicate the distribution of the loaded/unloaded leg amongst these men and women (to ensure that not all women were doing the right leg, and all men the left for example).
Line 70: did you do stratified randomization?
Line 74: “system” Please use more precise vocabulary. What type of system is it? walker, insoles, crutches, measurement device etc
Line 75: In standard English a "walker" is a walking frame that helps an individual walk, not something worn on the foot. The "walker" referred to in the SP Air Smart Walker is the model’s name but it refers to a foot brace which would be the more appropriate term.
Line 78: “single-use force measuring sole” what is the model/manufacturer?
Line 82: At what frequency did the data logger record the data?
Line 88: Do you know if participants used the visual cue as it requires the participants to look down at their foot, making the cue perhaps less used during community ambulation? Could be interesting and relevant to mention this in the discussion.
Line 99: What a 3-point-gait is needs to be slightly explained (e.g., "...to walk using a 3-point-gait pattern through the use of forearm crutches...")
Line 100: It's not clear to me what is happening. After being trained on how to walk in the 3-point pattern they then used a bathroom scale to get a feel of what 15-30 kg target zone feels like when doing partial weight bearing. This is done stationary correct? How many times did they do this? Just once? I find the term training to me implies more than once or more than one session, otherwise "familiarization session" might be a more appropriate term.
If I understood the protocol properly, in a second step the goal was to see if this target zone could be maintained while participants were walking and descending stairs. Correct? If yes, this is not very clear and I recommend the authors try to describe the protocol in simple but clear terms.
Line 101: Try to be consistent in terminology between participants vs subjects. Participants is the more commonly used and appropriate term.
Line 103: “Which leg the subject unloaded…” It seems confusing to put this information here. I assume this goes hand in hand with the 3-point -walk gait pattern training so should perhaps be mentioned there.
Line 104: “Analysis was performed first without biofeedback.” This information feels like it is coming too late. I would suggest specifying earlier that these previous steps were done without feedback and that only after this "baseline" recording the two groups were formed. As for the “analysis” this typically refers to the data analysis and seems odd to mention when you are still describing the protocol.
Line 107: Is control the same as the original run no? Nothing changed? If yes, perhaps just stating this clearly would help improve the understanding of the protocol and subsequent comparisons being made.
Line 111: Figure 2 is not cited in-text. Be sure all figures and tables are cited in-text.
Line 120: What normality tests were done? What groups were compared?
Results
Tables: Please report the exact p-values in the tables.
Line 148: Don’t need to repeat data that is already in the table.
Line 151: Isn’t “the side” the exception you mention right after? If yes, it doesn’t make sense to say here that there was no difference? If no, it is not clear what you are referring to here.
Line 154: As you present results for the left and right legs unloaded it would be helpful to know what the sex distribution of the unloaded legs were per side.
Discussion
Line 155: I feel the current content and structure of the discussion does not achieve the intended goals. The first portion of the discussion is better suited to the introduction while the second part is a bit disjointed and would need a proper interpretation of the results to be complete.
The discussion should interpret and explain the significance of the findings. It connects the results to a broader context in the field and addresses the aim of the study and it's hypotheses. The implications of the findings should also be mentioned.
This is typically structured as:
1) summary of key findings
2) Interpretation of results/implications/comparison with previous literature
3) limitations
4) future work
Line 164: “assessing movement” Movement or force? You talk about force before, so it seems important (but why it is important is never made clear) but then here you are talking about general movement assessment so does this also include force? Perhaps clarifying this would be beneficial to improve readability.
Line 170: “weight bearing behavior” what is the importance of this within rehabilitation? It would be good to explain this a bit to underscore the importance of having technology to help achieve this properly.
Line 156-186: I would say everything written in this section of the discussion is better suited to the introduction and would vastly improve the current introduction. What is written here far better highlights the background and reasoning justifying this study. Then instead of presenting the conclusion here you could highlight the gap in the literature and the aim of this study (if you move this all to the introduction).
Line 191: This isn't really a limitations and strengths section, just a general discussion. The discussion should be reformatted entirely.
Line 204: “…target zone has not yet been defined.” Have not been defined where? In the clinic? Because you define one here, don't you?
Line 205: “…are debatable…” If they are debateable, why did you choose them and why should the reader believe their validity?
Line 207: This paragraph seems very incomplete, perhaps join the ideas with those in the paragraph before?
Line 208: “…define this zone more precisely in patients’ clinical studies in the future…” What stopped you from defining them more precisely for this study?
Line 216: “…which makes the implementation of a common cross-company database appealing.” why would this be appealing? The reasoning behind this and support that this is needed isn't there.
Line 223: “like subjects” Please specify when you are referring to healthy or not otherwise it leads to confusion and decreased readability.
Conclusion
Line 234: “devices” You only look at one device so you can’t generalize across all devices.
Line 235: What makes it economical? This would be worth exploring in the discussion if it is a point you want to highlight in the conclusion.
Line 238: “…in an outpatient context between participants…” I am not convinced that such a broad generalization can be made from these results as you had a very restricted and controlled population so I don't know how this would transfer to the broader population.
Line 239: “feasibility” Since you didn’t explore this I don’t think you can make this last statement.
Comments on the Quality of English Language
Throughout the manuscript there is changing terminology, improperly used vocabulary, and a general structure that doesn’t align with the standard for a scientific paper. I would strongly recommend the authors re-work the writing in the manuscript to ensure that readability is improved. A native English speaker or professional proofreader should also have a detailed look though not only to ensure the correct terminology and vocabulary is used, but also the consistency throughout the text.
Round 2
Reviewer 1 Report
Comments and Suggestions for Authors
The manuscript has been satisfactorily revised. I recommend it be accepted.
Author Response
We would like to thank the reviewer for their time and effort. We greatly appreciate it.
Reviewer 3 Report
Comments and Suggestions for Authors
Thank you for the opportunity to re-review this work. I can see the effort the authors put into the revision, mainly in rewriting the introduction and discussion.
ABSTRACT:
Line 12: "..., employing a randomized…homogeneous subjects." I would remove this as it is anyways repeated below and it will help you gain some extra characters in your abstract.
"Line 16: How is "poor" defined? And how was compliance measured?"
Thank you for providing a definition of the term “poor.” However, the point that I was trying to make was that to ensure clarity and scientific rigor, it is important to contextualize this term with a reference value or benchmark. Without a specific reference, the term “poor” remains subjective and open to interpretation.
For instance, you could compare the compliance rate to a standard or guideline, such as a specific percentage threshold or a clinical recommendation. This would provide readers with a clear understanding of what constitutes “poor” compliance in this context.
"Line 18: If could be nice if you could report the exact p-value. Typically, the exact value is reported (P = .04) rather than expressing a statement of inequality (P < .05), unless P <.001."
I understand that the full manuscript contains the exact p-values. However, the abstract is intended to provide a concise yet comprehensive summary of the study’s key findings. Including the exact p-values in the abstract will allow readers to immediately understand the statistical significance of your results, enhancing the clarity and impact of your work.
Given that your target audience is familiar with interpreting p-values, this additional detail will not complicate the abstract but rather improve its precision and usefulness.
"Line 19: Hard to understand and interpret the results without knowing what the participants did."
It is still not clear to me in the abstract what tasks the participants did. As you mention overground walking and stairs in the results I would at minimum outline this before. If you remove the text, I suggested above it should give you enough characters to add something like “…target zone of 15-30kg during overground walking and going up and down stairs.”
INTRODUCTION:
Overall, I think the introduction has improved a lot. I think the ideas are more clearly presented with a clear motivation for the work and hypothesis. Some ideas don’t flow very smoothly and there are some jumps in logic that a proofreader could help with. For example, Line 32: “…which is why some degree of weight bearing is necessary [3-6]. An option for the assessment of gait parameters…” The flow between these ideas is not very smooth and it is quite a jump in logic.
Line 32: “…which is why some degree of weight bearing is necessary.” I think this sentence is misplaced as it contradicts the logic you are establishing. You are saying that partial weight bearing is done following surgery and that it is needed for wound healing and to mitigate the secondary loss of reduction. Then you are saying that this carries risks, but that this is why some partial weight bearing is necessary. Perhaps you can review this section. As a reader I feel like you are trying to say that the immobilization (not the partial weight bearing) is done to improve wound healing and mitigate risks, but that immobilization also has risks, which is why we need some partial weight bearing. However, this is not how it currently reads.
Line 73: What is meant by “…level of evidence to Ib”. Is this a standard? Perhaps a reference could help the reader find further information about this.
METHODS:
"Line 82: At what frequency did the data logger record the data?"
For the reply regarding the frequency of the logger I would just like to point out that this is standard information that could be found in the user manual or calculated directly from your data. It is not the responsibility of the reader to get this information from the manufacturer but for the authors to provide complete information as this could be relevant to interpretation of the data and results. Knowing the frequency is important for the reproducibility of the data, data quality (e.g., understand if it is low resolution), and methodological transparency.
"Line 88: Do you know if participants used the visual cue as it requires the participants to look down at their foot, making the cue perhaps less used during community ambulation? Could be interesting and relevant to mention this in the discussion."
You could add this point about the visual aspect of the biofeedback as a limitation of your study or elaborate on it in the discussion as suggested.
RESULTS:
I should have said it in the first revision round, but I notice now that the tables are quite unintuitive to understand. You currently have three tables baseline, control, and feedback each with the three types of walking you did. You are making comparisons across these tables with the p value representing the differences with the baseline walking. Wouldn’t it make more sense to group the data into tables per walking condition and show the baseline, control, and treatment group for that condition in each? That way you don’t have to scroll between three tables to see the values you are making your statistical comparison between.
DISCUSSION:
Line 214: This statement is misleading. This is not a conclusion from your study, but from the literature.
In the discussion it’s great that you now add some literature that has done similar work but it would also be important to highlight the differences between your work and theirs. Otherwise, the reader is left to wonder why your study is needed if these other studies found the same results. For example, for the study by Hershko et al., they also saw a benefit of an insole biofeedback system. What was different about their system, or limitations in their work that you addressed in your study?
I also don’t think you need to list the literature twice (in the first paragraph and again further down). You may consider outlining the main research question again before stating your findings. If you look in the literature or online I am sure you can find plenty of examples of how this first paragraph of the discussion is typically formulated.
Comments on the Quality of English LanguageThe authors have done a good job in preparing this manuscript, which is free of major grammatical errors and typos. The research presented is both valuable and well-executed. However, there is still room to enhance the clarity and overall flow of the writing, which would significantly improve the manuscript's readability.
In particular, the transitions between ideas and sections could be smoother, as some parts currently feel somewhat disjointed. Additionally, portions of the manuscript come across as more of a list of facts and references rather than a cohesive narrative. A more refined writing style would benefit these sections.
While I understand that hiring a professional proofreading service may not always be feasible in terms of budget or time, doing so could greatly enhance the manuscript. Beyond correcting errors, a professional proofreader could help restructure or rephrase certain passages, ultimately elevating the manuscript to a higher professional standard.
